# *PyEt* v1.3.1: a Python package for the estimation of potential evapotranspiration

Matevz Vremec[1,*], Raoul A. Collenteur[2,*], and Steffen Birk[1]

[*]These authors contributed equally to this work.
[1]Department of Earth Sciences, NAWI Graz Geocenter, University of Graz, Graz, Austria
[2]Eawag, Swiss Federal Institute of Aquatic Science and Technology, Department Water Resources and Drinking Water, Dübendorf, Switzerland

**Correspondence:** Matevz Vremec (matevz.vremec@uni-graz.at)

**Abstract.**

Evapotranspiration (ET) is a crucial flux of the hydrological water balance, commonly estimated using (semi-)empirical formulas. The estimated flux may strongly depend on the formula used, adding uncertainty to the outcomes of environmental studies using ET. Climate change may cause additional uncertainty, as the ET estimated by each formula may respond differently to changes in meteorological input data. To include the effects of model uncertainty and climate change, and facilitate the use of these formulas in a consistent, tested, and reproducible workflow, we present *PyEt*. *PyEt* is an open-source Python package for the estimation of daily potential evapotranspiration (PET) using available meteorological data. It allows the application of twenty different PET methods on both time series and gridded datasets. The majority of the implemented methods are benchmarked against literature values and tested with continuous integration to ensure the correctness of the implementation. This article provides an overview of *PyEt*'s capabilities, including the estimation of PET with twenty PET methods for station, and gridded data, a simple procedure for calibrating the empirical coefficients in the alternative PET methods, and estimation of PET under warming and elevated atmospheric $CO_2$ concentration. Further discussion on the advantages of using *PyEt* estimates as input for hydrological models, sensitivity/uncertainty analyses, and hind/forecasting studies, especially in data-scarce regions, is provided.

## 1   Introduction

Evaporation — the process by which water is converted from its liquid to vapor phase — is a central component of the global hydrological cycle (Katul et al., 2012). Evaporation has far-reaching impacts on both human societies and ecosystems (Oki and Kanae, 2006; Fisher et al., 2011). In the remainder of this paper, the term evapotranspiration (ET) is used to refer to the total evaporation flux from soil and water bodies (evaporation) and vegetated surfaces (transpiration; Allen et al., 1998; Dingman, 2015). Information about the magnitude of the ET flux is important across different geoscience disciplines: it assists in predicting irrigation demands and crop water requirements in agriculture, supports efficient water resources management, guides operational strategies in hydropower and meteorological studies, and plays a crucial role in ecological research and climate-change impact assessments. Given that climate change — through warming and elevated $CO_2$ concentrations — is set

to alter evapotranspiration, the need for its accurate estimation is of paramount importance, as it affects our understanding and assessment of past, present, and potential future impacts on ecosystem functioning (Milly and Dunne, 2016; Yang et al., 2019; Caretta et al., 2022).

Evapotranspiration can hardly be measured directly (Wang and Dickinson, 2012; Jensen and Allen, 2016), and is therefore commonly estimated using (semi-)empirical formulas from other, more easily obtained meteorological variables such as temperature, wind speed, and radiation. Over time, dozens of methods have been proposed and applied. Each of these methods generally results in slightly different estimates of evapotranspiration, depending on the methods and data used (Oudin et al., 2005; McMahon et al., 2013; Xu and Singh, 2000, 2001; Lemaitre-Basset et al., 2022). Most of these formulas estimate either the reference crop evapotranspiration ($ET_0$), which is ET from a reference surface or crop that is not short of water (Allen et al., 1998), or the potential evapotranspiration (PET), which is the maximum rate of ET that would occur given a sufficient water supply (Xiang et al., 2020). Potential evapotranspiration is determined by meteorological conditions, whereas water availability determines if actual evapotranspiration occurs at its potential rate (Jensen and Allen, 2016). Differences in the potential evapotranspiration estimate may cascade through a modeling chain and ultimately impact the results of a study. For example, Prudhomme and Williamson (2013), Lemaitre-Basset et al. (2022), and Bormann (2010) showed that the method used affects the results from hydrological climate change impact studies. Similarly, the estimation of water demand for efficient crop and irrigation management depends on potential evapotranspiration, and may thus be impacted by the methods used (Kumar et al., 2012).

To account for the structural uncertainty of the different PET models, it has been recommended to use multiple methods (Seiller and Anctil, 2016; Beven and Freer, 2001; Velázquez et al., 2013). Such an approach can help improve the understanding of the effect of model uncertainty on PET estimates in, for example, historical climate studies (Zhou et al., 2020; Dakhlaoui et al., 2020; Yang et al., 2019) and climate change impact studies (Bormann, 2010; Seiller and Anctil, 2016; Gharbia et al., 2018; Shi et al., 2020). Climate change impact studies often rely on climate projection data or global observational datasets, which are generally available in a gridded format (i.e., netCDF, GRIB), thus requiring tools that can efficiently process such data. Some of these datasets also contain only a limited set of observed or projected meteorological variables, requiring PET estimation methods that use fewer inputs. It may also be necessary to account for environmental variables that change over time and impact the evapotranspiration, such as vegetation changes and increases in atmospheric $CO_2$ concentrations (Fatichi et al., 2016; Ainsworth and Rogers, 2007; Vremec et al., 2022). Studies like those mentioned above, require software programs that 1) include multiple PET estimation methods, 2) be flexible in adjusting input parameters (e.g., empirical coefficients, crop data, and meteorological inputs), and 3) be applicable to both time series and gridded data, given the spatial nature of many of these studies.

Existing tools for calculating evapotranspiration, such as 'Evapotranspiration' in R (Guo et al., 2016), and "PyETo" (Richards, 2019) and 'pyfao56' in Python (Thorp, 2022), are primarily designed for station-based time-series data. This limits their applicability with gridded datasets. While Peterson et al. (2020) extended 'Evapotranspiration' to 'AWAPer' to process gridded data, its use is limited to Australia. For the large community of geoscientists working with Python, the number of available PET methods from existing packages is limited (3 for PyETo and 1 for pyfao56), compared to 21 methods in the R package. This

highlights a gap in the availability of a software for the estimation of multiple PET methods for both time-series and gridded

data, with the input parameter flexibility required for advanced studies on PET. Given the increasing need to understand and predict environmental changes accurately across the globe, the availability of such software is of paramount importance for the geoscience community.

Opportunities also exist to further align these tools with the FAIR standards of findability, accessibility, interoperability, and reusability for research software, crucial importance for the credibility and reproducibility of scientific studies (Barker et al.,

2022). This involves improving methodological testing through continuous integration, inclusion of additional alternative PET methods, and enabling more flexibility in adjusting internal empirical coefficients. Such enhancements not only adhere to best practices in software development, but also broaden the scope and applicability of these tools in diverse geoscientific contexts. The refinement and development of evapotranspiration estimation tools that fully embrace the FAIR principles are therefore crucial steps toward advancing the field, ensuring more reliable and comprehensive research outcomes in the face of evolving

scientific needs (Wood et al., 1998; DeJonge and Thorp, 2017).

In this paper we introduce *PyEt*, an open-source Python package for the estimation of potential evapotranspiration. The aim of *PyEt* is to provide researchers and practitioners with a wide variety of tested, documented, and flexible Python functions that support multiple PET methods for both station and gridded data. All methods have a common application programming interface, allowing users to easily test different PET models for their application and, if desired, address structural uncertainty

and changing conditions. The majority of the implemented methods are benchmarked against literature values and tested with continuous integration to ensure the correctness of the implementation. Allowing different types of input data, *PyEt* is also applicable in regions with sparsely distributed measurement stations, where standard meteorological data (e.g., wind, relative humidity) are often unavailable. The software is available under MIT-license from the Python Package Index (PyPI) (Vremec and Collenteur, 2022), and developed as a community project on GitHub (www.github.com/pyet-org/PyEt).

The remainder of this paper is structured as follows. In the next section, the software design, capabilities, and benchmarking tests are described. The third section introduces the software through four examples, showing potential future users how to apply *PyEt* in real-world applications. These examples concentrate on addressing practical problems commonly faced by geoscientists in their daily work. The fourth section discusses future potential applications of *PyEt*, and how we think it can help the scientific community improve the estimation of potential evapotranspiration. In the fifth and final section, conclusions

and future plans are outlined.

## 2    PyEt Python Package

### 2.1    Software design

The basic design principle for *PyEt* was to build a software that is intuitive and easy-to-use by novice users with little programming experience, yet flexible enough to allow advanced users to perform more complex analyses. The software uses a

modular design, with formulas shared by different PET methods implemented as a single function. This reduces the amount of code and makes it easier to maintain the software and implement new methods. All the PET methods are intended to work

with the minimum input data required by the PET models (e.g., radiation, temperature), but also allow more user input if such data is available and allowed by the PET method (e.g., humidity, surface resistance in the Penman-Monteith model). Utility functions are available to the user or are called internally to compute unavailable variables (e.g., solar radiation from latitude value). Moreover, the constants in the empirical PET formulas (e.g., the Stefan Boltzmann constant) are function arguments with default values from the literature, which may also be changed by the user to adapt the empirical relationship to another region. Finally, the available methods should work for both station (1D) and gridded data (2D/3D).

*PyEt* is part of the wider Python ecosystem, and depends on three widely used and well-developed Python packages from the scientific Python stack: Numpy (Harris et al., 2020), Pandas (McKinney, 201), and Xarray (Hoyer and Hamman, 2017). The input and output data of *PyEt* are formatted as time series data in `Pandas.Series` or `Xarray.DataArrays`, which allows using all the Pandas or Xarray functions on the data (Harris et al., 2020; McKinney, 201; Hoyer and Hamman, 2017). These functions include gap-filling and selection functions for interpolation, resampling, clustering, and many more. Being part of a wider ecosystem, users can leverage other Python packages for visualization (e.g., Matplotlib (Hunter, 2007), MetPy (May et al., 2022)) and optimization and uncertainty analyses (Scipy (Virtanen et al., 2020), SpotPy (Houska et al., 2015)).

The software is hosted and developed on the GitHub platform, and distributed under MIT-license through the Python Packaging Index (PyPI). Documentation and example applications are available on a dedicated ReadTheDocs website (http://pyet.readthedocs.io). The documentation for individual methods is also directly available in Python from the documentation strings. Each release of *PyEt* is automatically stored in the Zenodo repository and assigned a Digital Object Identifier (DOI). As such, *PyEt* complies with many of the recommendations for good research software development as given in, for example, Hutton et al. (2016) and the FAIR4RS (FAIR for Research Software) principles (Barker et al., 2022). The scripts or the Jupyter notebooks used to apply *PyEt* improve the reproducibility and provide a transparent report of the entire calculation process (Kluyver et al., 2016).

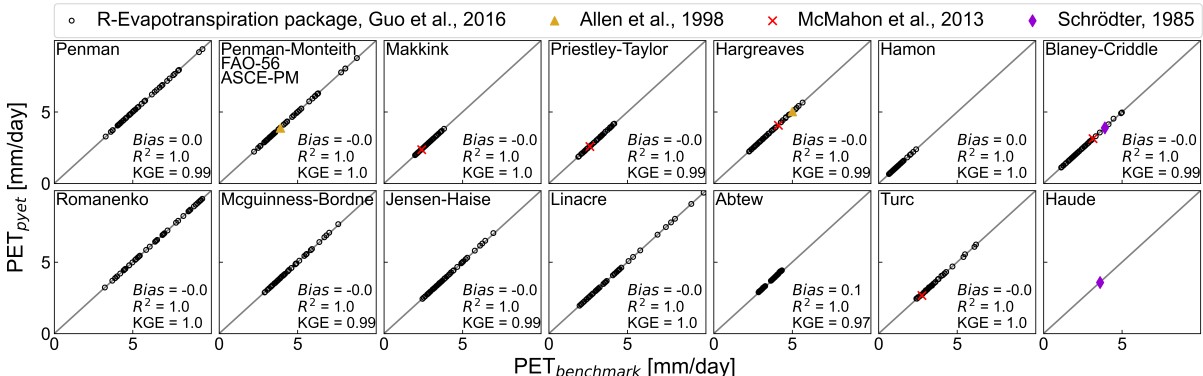

**Figure 1.** Scatter plots showing estimated PET with *PyEt* against PET values estimated with the R package *Evapotranspiration* from Guo et al. (2016), and literature values from Allen et al. (1998), McMahon et al. (2013), and Schrödter (1985).

## 2.2 Implemented methods and benchmarking

Twenty methods are currently implemented in *PyEt* for the estimation of daily potential evapotranspiration. Apart from the Penman-Monteith method, which is considered as the standard by the Food and Agriculture Organization (FAO) (Allen et al., 1998) and the World Meteorological Organization (WMO), multiple alternative methods are also available in *PyEt*. An overview of these methods and the required input data is provided in Table 1. Depending on the method, different (amounts of) input data are required to compute the potential evapotranspiration. It is often also possible to provide different input data to the same method (e.g., the average or the minimum and maximum daily temperatures), or even that some input data is optional (as described in the footnotes of Table 1). In the case of optional input data, utility functions are used internally to estimate that data. In the example of the Penman-Monteith method, solar radiation does not necessarily need to be provided by the user and can be estimated from the latitude and actual duration of sunshine hours instead.

The *PyEt* project is intended to be used by a wide community, and any errors in the code may have consequences for other studies applying *PyEt* to obtain PET estimates. Special attention was therefore paid to benchmark the available methods to published literature values and data from well-known research and meteorological institutes (Allen et al., 1998; McMahon et al., 2013; Schrödter, 1985; Walter et al., 2000). These benchmarks are also implemented in the continuous integration and tested using the *unittest* testing framework (unittest, 2022). This ensures that the benchmarks are satisfied each time the software is updated in the future. New methods added to *PyEt* will be required to be accompanied by the appropriate benchmark data and tests. Figure 1 shows the results for each benchmarked method, indicating that the PET estimates from all these methods are equal to the benchmark values (i.e., all values are on the 1:1 line). Despite our best efforts, we acknowledge here that four methods have not (yet) been benchmarked due to a lack of appropriate data.

In various sections of the manuscript, the performance of PET estimation methods are evaluated using three key performance metrics: model bias (mm/day), the coefficient of determination (-), and the Kling-Gupta Efficiency (KGE, -) (Gupta et al., 2009). These metrics enabled comparisons between benchmark PET values from literature and those estimated with PyEt, as well as between PET values derived from alternative models and the Penman-Monteith method, both before and after calibration. The Python implementations of the package SpotPy (Houska et al., 2015) were used for each performance metric, the formulas of which are detailed in the Appendix A1.

## 2.3 Performance

The computational efficiency of various PET methods was assessed by examining the computation times in relation to the time series length and the number of cells in a Xarray dataset. Computation time was evaluated by running all models on a benchmark configuration (with time series of varying lengths) using a 12th Gen Intel Core i7-1255U processor with 10 cores and 12 logical processors. All `Xarray.DataArrays` cover a period of 1 year, while the spatial resolution changes. This comparison highlights the trade-offs between computational complexity and data size, but also demonstrates the performance of the methods.

**Table 1.** Data requirements for different PET or ET$_0$ models, the corresponding *PyEt* function, and if benchmarking of the method was performed. The references include both the original publications of the models and the manuscripts from which the equations were taken: [1] McMahon et al. (2013), [2] Oudin et al. (2005) [3] Xu and Singh (2001), [4] Ansorge and Beran (2019); Rosenberry et al. (2004), [5] Schrödter (1985), [6] Schiff (1975), [7] Jensen and Allen (2016), [8] Xu and Singh (2000)

| Method name[0] | *PyEt* function | Climate data | | | | Location | | Bench. | Literature |
|---|---|---|---|---|---|---|---|---|---|
| | | $T$ | $RH$ | $R$ | $u_2$ | Lat. | El. | | |
| Penman-Monteith | *pm* | ✓[a] | ✓[b,c] | ✓[d] | ✓ | ✓[d] | ✓[e] | ✓ | Monteith (1965) |
| ET$_0$: ASCE-PM | *pm_asce* | ✓[a] | ✓[b,c] | ✓[d] | ✓ | ✓[d] | ✓[e] | ✓ | Walter et al. (2000) |
| ET$_0$: FAO-56 | *pm_fao56* | ✓[a] | ✓[b,c] | ✓[d] | ✓ | ✓[d] | ✓[e] | ✓ | Allen et al. (1998) |
| Penman | *penman* | ✓[a] | ✓[b,c] | ✓[d] | ✓ | ✓[d] | ✓[e] | ✓ | Penman (1948) |
| Priestley-Taylor | *priestley_taylor* | ✓ | ✓[h] | ✓[h] | - | ✓[h] | ✓[e] | ✓ | Priestley and Taylor (1972) |
| Kimberly-Penman | *kimberly_penman* | ✓[a] | ✓[b,c] | ✓[d] | ✓ | ✓[d] | ✓[e] | - | Wright (1982) |
| Thom-Oliver | *thom_oliver* | ✓[a] | ✓[b,c] | ✓[d] | ✓ | ✓[d] | ✓[e] | - | Thom and Oliver (1977) |
| Blaney–Criddle | *blaney_criddle* | ✓ | -[i] | -[i] | -[i] | ✓ | - | ✓ | Blaney and others (1952), [1, 3, 5] |
| Hamon | *hamon* | ✓ | - | - | - | ✓ | - | ✓ | Hamon (1963), [2, 4] |
| Romanenko | *romanenko* | ✓ | ✓ | - | - | - | - | ✓ | Romanenko (1961), [3] |
| Linacre | *linacre* | ✓[j] | - | - | - | - | ✓ | ✓ | Linacre (1977), [3] |
| Haude | *haude* | ✓ | ✓[k] | - | - | - | - | ✓ | Haude (1955), [6] |
| Turc | *turc* | ✓ | ✓ | ✓ | - | - | - | ✓ | Turc (1961), [8] |
| Jensen–Haise | *jensen_haise* | ✓ | - | ✓[l] | - | ✓[l] | - | ✓ | Jensen and Haise (1963), [2, 7] |
| McGuinness–Bordne | *mcguinness_bordne* | ✓ | - | - | - | ✓ | - | ✓ | McGuinness and Bordne (1972), [8] |
| Hargreaves | *hargreaves* | ✓[m] | - | - | - | ✓ | - | ✓ | Hargreaves and Samani (1982), [1, 7] |
| ET$_0$: FAO-24 | *fao_24* | ✓ | ✓ | ✓ | ✓ | - | ✓[e] | - | Jensen et al. (1990) |
| ET$_0$: Abtew | *abtew* | ✓ | - | ✓ | - | - | - | ✓ | Abtew (1996), [8] |
| Makkink | *makkink* | ✓ | - | ✓ | - | - | ✓[e] | ✓ | Makkink (1957), [1] |
| Oudin | *oudin* | ✓ | - | - | - | ✓ | - | - | Oudin et al. (2005) |

[0] The corresponding literature to each method is provided in the Appendix A1. [a] $T_{max}$ and $T_{min}$ can also be provided. [b] $RH_{max}$ and $RH_{min}$ can also be provided. [c] If actual vapor pressure is provided, RH is not needed. [d] Input for radiation can be (1) Net radiation, (2) solar radiation or (3) sunshine hours. If (1), then latitude is not needed. If (1, 3) latitude and elevation is needed. [e] One must provide either the atmospheric pressure or elevation. [f] The PM method can be used to estimate potential crop evapotranspiration, if leaf area index or crop height data is available. [g] The effect of $CO_2$ on stomatal resistance can be included using the formulation of Yang et al. (2019). [h] If net radiation is provided, RH and Lat are not needed. [i] If method==2, $u_2$, $RH_{min}$ and sunshine hours are required. [j] Additional input of $T_{max}$ and $T_{min}$, or $T_{dew}$. [k] Input can be $RH$ or actual vapor pressure. [l] If method==1, latitude is needed instead of $R_s$. [m] $T_{max}$ and $T_{min}$ also needed.

Figure 2 demonstrates that all *PyEt* methods maintain computational times below 1 second for time series data with lengths ranging from 10 to 10,000 days. For multidimensional data, the computation time does not exceed 10 seconds for larger `Xarray.DataArrays` up to 100,000 cells. Moreover, the results in Figure 2 show that the problem scales well, and does not take proportionally more time for larger data sets. Notably, the Penman-Monteith and Priestley-Taylor methods exhib-

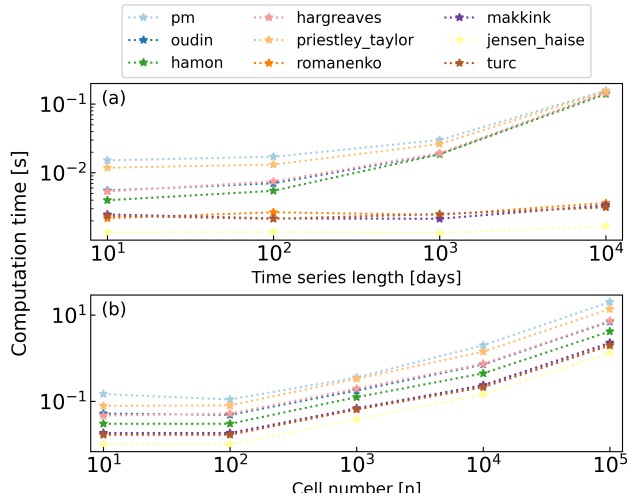

**Figure 2.** Computational efficiency of different PET methods:This figure shows the comparative processing times of different PET estimation methods regarding the length of the time series (a) and the size of the Xarray data (b).

ited the largest processing times, whereas methods like Jensen-Haise, Turc, Makkink, and Romanenko were faster. Future
improvements will aim to increase this efficiency, in particular to support faster calculations in large-scale global studies.

## 3 Example use cases

Below, four use cases of *PyEt* are presented to illustrate how the software can be used. The first example shows how to efficiently compute different potential evapotranspiration estimates using 20 various methods for station data. This example also illustrates how to use *PyEt* in general. The second example illustrates how to provide 3D estimates of PET using 3 different
methods and gridded Xarray data. The third example shows how to calibrate different PET methods to local conditions and use the calibrated formula for hindcasting. The fourth and final example illustrates a workflow to account for the effects of warming and elevated $CO_2$ in climate change impact studies. The source code for these and other examples can be found in a Zenodo repository related to this paper (Matevz Vremec and Raoul Collenteur, 2024).

### 3.1 Example 1: Estimation of PET from station data

In this example, potential evapotranspiration is estimated for the town of De Bilt in The Netherlands using data provided by the Royal Netherlands Meteorological Institute (KNMI). The reference method used by the KNMI for the estimation of potential evapotranspiration is the Makkink method, also implemented in *PyEt*. The PET computed with the Makkink method is compared to the PET values from all other methods in *PyEt*. Several steps are taken in a Python script to estimate PET. The code implementing these steps is shown in the code example below. *PyEt* provides a convenient way to compute the PET with
all available methods, *pyet.calculate_all()*:

1. Import the necessary Python packages.

```python
import pandas as pd
import pyet
```

2. Load the meteorological data.

```python
meteo = pd.read_csv("meteo.csv",
    index_col=0, parse_dates=True)
```

3. Determine the necessary input data for the PET model.

```python
tmean, tmax, tmin, rh, rs, wind, \
pet_knmi = (meteo[col] for col in
            meteo.columns)
lat = 0.91  # latitude
elev = 4  # elevation
```

4. Estimate the potential evapotranspiration with all methods or the method of choice.

```python
pet_df = pyet.calculate_all(tmean,
    wind, rs, elev, lat, tmax, tmin, rh)
pet_mak = pyet.makkink(tmean, rs,
    elevation=elev)
```

5. Visualize and analyze the results.

```python
pet_df.plot()
pet_df.boxplot()
pet_df.cumsum().plot()
```

The results from this analysis are shown in Figure 3. From these visualizations, it is clear that the potential evapotranspiration depends on the chosen method. This can accumulate up to a 35% deviation of the estimated annual flux from the mean in this example. Such substantial differences between the estimated fluxes motivate the use of multiple methods (ensemble modelling) (Beven and Freer, 2001; Krueger et al., 2010; Shi et al., 2020; Oudin et al., 2005). This example showed how *PyEt* can be used efficiently for this task.

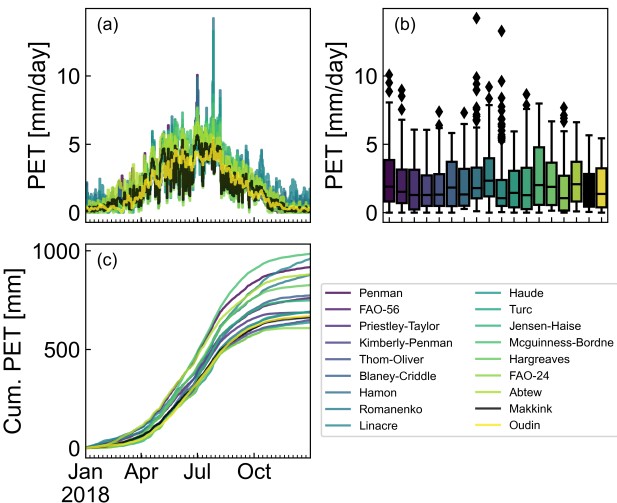

**Figure 3.** Potential evapotranspiration estimates for the year 2018 computed with all available PET methods, plotted as (a) time series, (b) box plots, (c) cumulative PET.

## 3.2 Example 2: Estimate PET for gridded data

Gridded 3-dimensional data (x, y, and t) obtained from satellites, radar imagery, or post-processed products is rapidly becoming widely available. More and more public data sets exist with global PET estimates at 0.1 degree resolution (e.g., Martens et al., 2017; Xie et al., 2022). *PyEt* also supports such gridded data, as illustrated here for the E-OBS gridded data set (Cornes et al., 2018) for Europe. The application of *PyEt* on gridded datasets is illustrated for the FAO-56, Makkink, and Hargreaves method. `Xarray.DataArrays` are used as input data instead of `Pandas.Series`. *PyEt* methods will return the same data type, a `Xarray.DataArray`. The workflow is comparable to that in the first example, except that now the individual PET methods are used.

The results for the three methods and three time steps are shown in Figure 4. These again show that, depending on the PET method, results may differ, also spatially. Looking more closely at Figure 4, we can observe that the FAO-56 and Makkink method do not compute PET in eastern parts of Europe. The data do not include relative humidity and solar radiation for these areas, and thus PET cannot be computed using the FAO-56 or Makkink method. If NaN (not-a-number) values are present in the required input data for a *PyEt* method, the method also returns a NaN value. The Hargreaves method, on the other hand, does not require solar radiation or relative humidity data. It can therefore be used to compute PET in the eastern parts of Europe. This example showed how *PyEt* can be applied to estimate PET using gridded data, and demonstrated the benefits of using alternative PET methods when data such as radiation or relative humidity are missing.

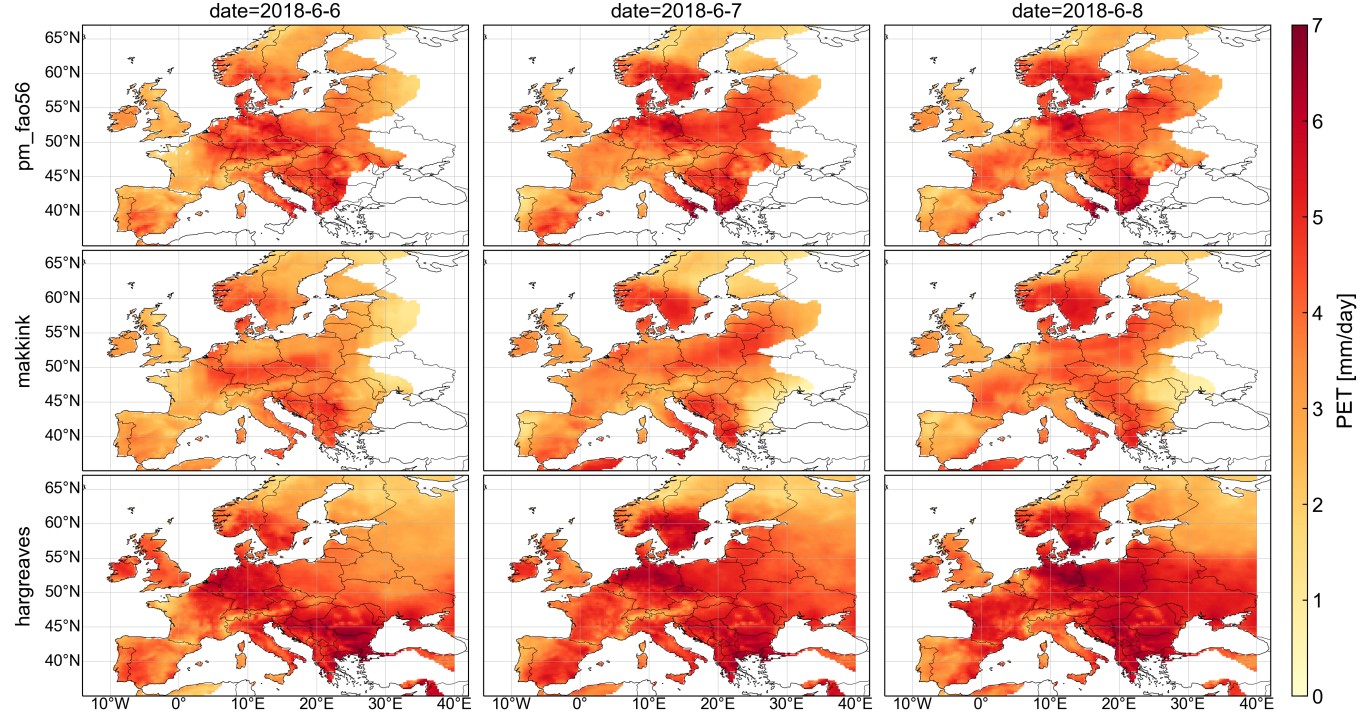

**Figure 4.** Daily PET estimates for Europe from 2018-6-6 to 2018-6-8 using meteorological data obtained from the E-OBS dataset (Cornes et al., 2018).

## 3.3 Example 3: Calibration of PET models

The available input data often does not suffice to compute potential evapotranspiration with the Penman-Monteith equation. This can be the case in data-scarce regions or time periods, or when using historical data or data from climate models. In such cases, alternative PET methods can be calibrated to the estimates obtained from the Penman-Monteith equation for a period when sufficient data is available. The calibrated method can then be used to estimate PET in periods of data scarcity. As concluded by several authors (Jensen and Allen, 2016; Valipour, 2015; Yang et al., 2021; Dlouhá et al., 2021), calibration of alternative models is often crucial to ensure that the model fits the regional climate. In this example, it is shown how the calibration of temperature-based PET models affects the model uncertainty for studies focusing on current and past climates.

The approach is illustrated for the town of Graz, Austria, where the input data required for Penman-Monteith are only available from 2000 to 2021. Imagine, however, that for our study we also need potential evapotranspiration data for the period 1961 to 2021, but only temperature data is available (e.g., from the Spartacus temperature dataset (Hiebl and Frei, 2016)). Several steps are taken to calibrate the following five temperature-based methods: Oudin, Hargreaves, McGuiness-Bordne, Hamon, and Blaney-Criddle. First, the PET for the period 2000-2021 is computed using the Penman-Monteith equation. In the second step, the coefficients of the temperature-based PET equations are estimated by calibrating the estimated PET from

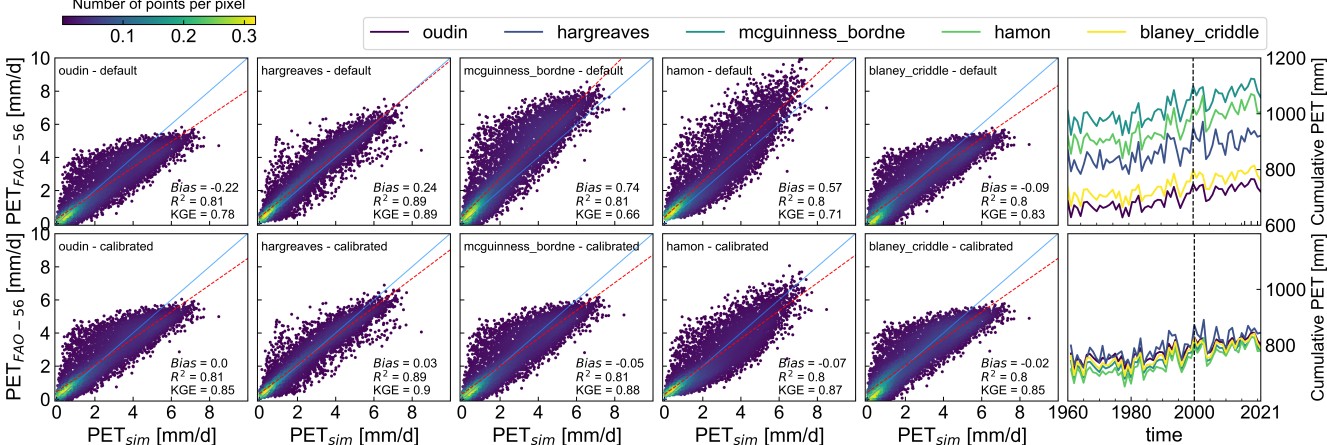

**Figure 5.** Density scatter plots comparing simulated and observed (FAO-56) PET for uncalibrated (row 1) and calibrated models (row 2). The performance of the calibrated models is evaluated using the model bias in mm/day (*Bias*), the coefficient of determination ($R^2$), and the Kling-Gupta Efficiency (KGE). The last column shows the annual PET sums for the period 1961-2021 using uncalibrated (row 1) and calibrated models (row 2).

temperature-based methods to the Penman-Monteith PET. Calibration is done by minimizing the sum of the squared residuals between these two PET estimates, using SciPy's (Virtanen et al., 2020) *least_squares* method. In the third and final step, these calibrated coefficients are used to estimate the PET for the period 1961-2021.

Figure 5 shows the computed PET with default (row 1) and the calibrated coefficients (row 2). The model bias (mm/day) and the Kling-Gupta Efficiency between simulated and observed (Penman-Monteith) PET show an improved model fit for all methods after calibration. The use of calibrated methods reduces the model bias, which is visually illustrated by the annual PET flux (composed of daily values) in the last column of Figure 5. Using the Spartacus temperature dataset (Hiebl and Frei, 2016), PET can now be estimated back to 1961 using the calibrated alternative PET methods.

**3.4 Example 4: The effect of $CO_2$ on future PET estimates**

In this example, it is shown how to account for changing environmental conditions affecting the PET flux when modelling the effects of climate change. Under a warmer and $CO_2$ richer future (Caretta et al., 2022), potential evapotranspiration tends to increase with increasing temperature (and vapor pressure deficit). A reduction in PET is expected under elevated $CO_2$ due to an increased stomatal resistance (Field et al., 1995; Ainsworth and Rogers, 2007). The increase in $CO_2$ is still commonly ignored

in PET models employed for climate change studies, although excluding its stomatal effect may lead to an overestimation of PET (Kingston et al., 2009; Milly and Dunne, 2016; Vremec et al., 2022; Riedel et al., 2023). The effect of temperature increases on PET can be easily modelled with all available PET methods, as temperature is an input for all methods. The $CO_2$ stomatal effect, however, can only be directly accounted for with the Penman-Monteith method (Liu et al., 2022). Using a $CO_2$-dependent stomatal resistance model implemented in *PyEt* (Yang et al., 2019), the effect of elevated $CO_2$ on stomatal

resistance can be considered (see Eq. A2). When calculating PET with alternative methods, Kruijt et al. (2008) and Trnka et al. (2014) argued that an adjustment factor for the atmospheric $CO_2$ concentration ($f_{CO_2}$) can be used to account for the effect of elevated $CO_2$ concentrations on PET. The scaling factor can be obtained from literature values (Kruijt et al., 2008; Trnka et al., 2014). Alternatively, the factor can be calibrated using the Penman-Monteith equation together with the $CO_2$-dependent stomatal resistance model (Eq. A1) to match the local climate and vegetation:

$$\mathrm{PET}_{CO_2} = f_{CO_2}\mathrm{PET}_{300}$$

$$= (1 + S_{\mathrm{PET}_{CO_2}}(CO_2 - 300))\mathrm{PET}_{300} \tag{1}$$

where $S_{PET_{CO_2}}$ is the relative sensitivity of PET to $CO_2$, $\mathrm{PET}_{300}$ is the computed Penman-Monteith estimate at 300ppm [$CO_2$] (preindustrial concentration), while $\mathrm{PET}_{CO_2}$ is the computed Penman-Monteith estimate under elevated $CO_2$ concentration (Yang et al., 2019). Such relationships can be easily implemented in *PyEt*, and $f_{CO_2}$ can be obtained by calculating $\mathrm{PET}_{300}$ and $\mathrm{PET}_{CO_2}$ with the Penman-Monteith equation (Eq. A1) at ambient and elevated $CO_2$ concentration, respectively.

Building on the previous example, the Graz study area served as a practical example to demonstrate the application of the calibrated models in assessing the impact of warming and elevated $CO_2$ concentration on PET based on the projected increase in temperature and $CO_2$ concentration from the representative concentration pathways (RCPs) (Van Vuuren et al., 2011). Daily PET was calculated for each RCP scenario (2.6, 4.5, 6.0, and 8.5) by adding the projected increase in temperature and $CO_2$ concentration to the existing data for 2020-2021. Figure 6 shows the increase in the average annual PET (aggregated from daily values) under warming and elevated $CO_2$ concentrations according to the RCP scenarios. In figure 6c, the effects of elevated

$CO_2$ concentration on PET were neglected, and only increases in temperature were considered. Similar to Milly and Dunne (2016), Yang et al. (2019), and Vremec et al. (2022), this example shows that neglecting the effect of elevated $CO_2$ on PET (Fig, 6-c) can lead to overestimation of PET under future conditions.

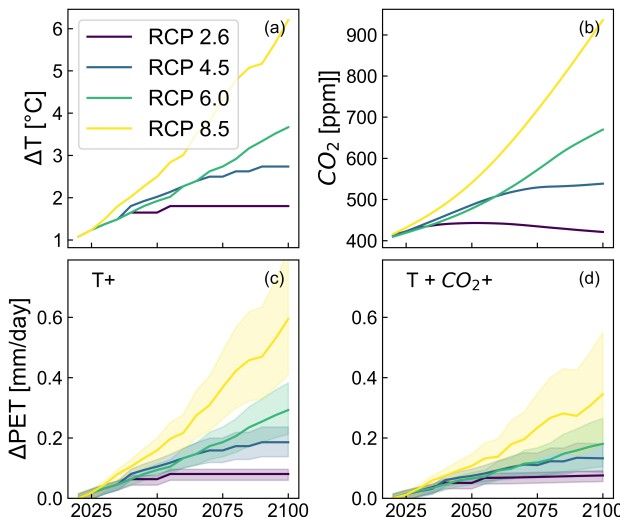

**Figure 6.** Projected increase in temperature (6-a) and atmospheric $CO_2$ concentration (6-b) under the RCP scenarios, and calculated increase in the average annual PET with warming (6-c), and PET with warming and elevated $CO_2$ concentration (6-d). The uncertainty bounds represent the 5th-95th percentile of the PET model ensemble.

## 4    Discussion

### 4.1    Improved handling of PET in scientific studies

Evapotranspiration data from lysimeter or eddy correlation measurements (Pastorello et al., 2020) are rare and, if available at all, only locally available for relatively short time periods. Thus, there is a widespread need to estimate evapotranspiration from more readily available meteorological data using (semi-)empirical approaches. In general, these approaches follow three steps, as outlined for example by Allen et al. (1998). Firstly, the potential evapotranspiration of a reference surface (hence reference evapotranspiration) is estimated using meteorological data. Secondly, a crop coefficient may be applied to transform the reference evapotranspiration into the potential crop evapotranspiration. Thirdly, a soil-water balance approach is used to account for reduced actual evapotranspiration if the soil-water storage is depleted. *PyEt* is designed to perform the first two steps. It can be easily complemented by soil-water balance approaches to calculate actual evapotranspiration. Hydrological models, however, often use PET directly as input.

Rainfall-runoff models represent one type of hydrological model where PET is commonly used as an input, either as gridded data in distributive models, or as a spatially aggregated values in lumped-parameter models. Some studies (e.g., Andréassian et al., 2004; Oudin et al., 2005; Sperna Weiland et al., 2012) found that PET had little impact on the performance of such models, and thus advocated the use of simplistic PET models. However, Jayathilake and Smith (2021) found that model performance was clearly sensitive to PET at sites where evapotranspiration was water limited. More importantly, the choice of the PET model has been shown to affect the results of hydrological projections in climate change impact assessments (Kay and

Davies, 2008; Seiller and Anctil, 2016; Dallaire et al., 2021; Lemaitre-Basset et al., 2022)). PET is expected to be even more influential in the assessment of groundwater recharge (e.g., Bakundukize et al., 2011) and crop water demands (e.g., Webber et al., 2016), which – compared to runoff – are more directly linked to evapotranspiration. Thus, the selection of appropriate PET models needs to account for the research context and variable of interest.

Bormann (2010) found that PET models that are based on the same or similar climate variables exhibit different sensitivity to observed climate change. This finding suggests that appropriate PET models need to be specifically selected for the given region of interest. Guo et al. (2017) provides pointers to examining which variables are likely to be the most important for a particular location. For more detailed insights into the application of individual PET methods across various climates, refer to Table A2 and the studies by Allen et al. (1998); McMahon et al. (2013); Jensen and Allen (2016); Yang et al. (2021); Pimentel et al. (2023). The comparison of PET estimates for Europe shown in Figure 4 illustrates the spatial variability of differences in PET estimates obtained from different methods; as can be seen, the magnitude and pattern of PET estimates are similar in some regions (e.g., Scandinavia) but differ more strongly in others (e.g., Southeast Europe).

As indicated above, the performance of PET models may vary depending on the region considered. Approaches that were found applicable in one region may perform less well in other regions. In this case, PET models can be calibrated to a reference data set by adjustment of the coefficients in the model equation, as shown in the third example. The reference data set can either be observed evapotranspiration (e.g., from lysimeters) or PET obtained from a model considered to be reliable. This has been illustrated by Example 3, where the coefficients of temperature-based models were adjusted to achieve the best fit to the Penman-Monteith model. This approach can also be used to obtain consistent spatial distributions of PET. As shown in Example 2 (Figure 4), the limited data availability for Eastern Europe did not allow the application of the FAO-56 or Makkink method, while sufficient data was available for the Hargreaves method. Thus, one may consider calibrating the latter to one of the former methods where these are applicable, and only then apply it to obtain estimates for the entire region. For a more advanced calibration procedure, see for example Haslinger and Bartsch (2016).

Often, the range of PET models that can potentially be employed is pre-determined by data availability. This may be the case if historical records of climate data are to be used for the PET estimation, for example, as many weather stations do not measure all climate variables included in the Penman-Monteith equation. Yet, this is also often the case in assessments of hydrological impacts of climate change if projected climate variables have high uncertainty. Lai et al. (2022), for example, concluded that the high uncertainty of wind speed projected in complex terrain may increase the uncertainty in PET, whereas air temperature and solar radiation have low uncertainty and thus should be the parameters preferred in the PET model. Given the climate variables for which data is available, Table 1 can be used to identify the PET models that come into consideration. However, it is advised to evaluate the assumptions and limitations of the individual methods regarding their applicability in the given case. Please refer to the comments in Table A2 and the references in Table 1 for this purpose. We generally recommend applying all models that have been identified as suitable (PET model ensemble), but the purpose and specific implementation of such a multi-model approach will depend on the research context. Example 4 (section 3.4) illustrated how PET model ensembles can be used to include model uncertainties in PET projections under warming and elevated atmospheric $CO_2$ concentration. Since the latter effect is frequently excluded in hydrological projections, Milly and Dunne (2016) and Yang et al. (2019) advocate the

inclusion of the effect of elevated $CO_2$ on stomatal resistance when estimating PET under warming and elevated atmospheric $CO_2$ concentrations.

To evaluate the performance of the estimated PET against observed values or other PET methods, performance metrics are employed. In the manuscript, the model bias and coefficient of determination ($R^2$) are used to provide readers and practitioners with a clear and concise assessment of the overall deviation in flux and the percentage of the variability explained by the model (Onyutha, 2024). Additionally, the Kling-Gupta Efficiency (KGE) (Gupta et al., 2009) is used, offering a comprehensive view by incorporating correlation, variability, and bias into a single statistic. KGE is particularly valued in hydrological modeling, as it addresses some limitations of the $R^2$ by providing a differentiated perspective on model performance. The choice of the most appropriate performance metric is important to reduce model uncertainty, as it affects the judgement of the performance of a method. For further guidance on the evaluation of various performance metrics, readers are referred to Krause et al. (2005); Onyutha (2024).

To improve reliability and efficiency in estimating PET, it is crucial to use a reproducible workflow. Scripts provide an efficient way to document the modelling process and are an important step towards full reproducibility. As shown in the examples, Jupyter Notebooks (Kluyver et al., 2016) provide a solution for publishing code, results, and explanations in a single document. As such, the presented package and its application in this paper are in line with the steps suggested by Hutton et al. (2016) to improve reproducibility in hydrological studies. To speed up adaptation of the methods and allow a faster transfer between research teams, formal procedures such as benchmarking (e.g., Maxwell et al., 2014) can help to ensure confidence in key complex codes.

## 4.2 Building the PyEt community and outlook

As a community project, the success of *PyEt* depends on the uptake from and interaction with the community. This, in turn, depends on the ease of use and the trust in the project. Emphasis was put on designing a user-friendly, well-documented software, including various user examples, and extensive benchmark testing using continuous integration. Since the initial launch of *PyEt*, the package has already seen a good community uptake. Apart from applications of the software in projects related to the Authors, which include estimating PET under conditions of warming and elevated $CO_2$ concentrations, assessing potential crop evapotranspiration, and providing inputs for hydrological models (e.g., Vremec et al., 2022; Forstner et al., 2022; Collenteur et al., 2023; Jemeljanova et al., 2023), *PyEt* has also been independently used by other researchers. These studies have used *PyEt* for various purposes: integrating PET estimates with machine learning models for enhanced analytical capabilities (Vaz et al., 2022; Kajári et al., 2024), and combining them with software for computing groundwater recharge and the water balance (Hassanzadeh et al., 2024). The software has also played a role in evaluating hydroclimatic changes and generating regional and global PET estimates (e.g., Tercini and Mello Júnior, 2023; Aguayo et al., 2024; Ha et al., 2024). The quick uptake of the software by the community confirms the need for this software.

The primary channel for communication with the *PyEt* community is GitHub, which provides several options for discussions, tracking code issues, and code development. Users are encouraged to ask questions in GitHub discussions and to report potential issues, suggest improvements, and feature requests via the GitHub issue tracker. As a community project, we plan to

continue to improve the existing code and develop new capabilities based on feedback and with help from the community. An example of developments that are currently underway is the adaptation of the current methods to also work for hourly data, allowing the estimation of hourly PET. Other future work will focus on improvements in usability and the inclusion of other alternative methods.

## 5 Conclusions

This paper introduced *PyEt*, a Python package for the estimation of daily potential evapotranspiration (PET). The package enables the inclusion of model uncertainty and climate change in the PET estimation in a consistent, tested, and reproducible environment. With *PyEt*, PET can be estimated using 20 different methods with just a few lines of Python code. Unlike existing tools for PET calculation, which are designed for station-based time series, *PyEt* can also be applied to gridded (3D) data sets. This is of great practical relevance, particularly in climate impact studies, where gridded data sets are often used. The examples described in this paper illustrate how *PyEt* can be used in geoscientific studies to (1) facilitate the characterization of model uncertainty using a multimodel approach (model ensembles); (2) calibrate PET models and apply them in data-scarce regions and time periods; (3) include the effects of warming and elevated atmospheric $CO_2$ concentrations. The use of Python scripts and Jupyter Notebooks ensure reproducibility and provides a transparent report of the PET computation process. We believe that *PyEt* will help improve the handling of PET and allow a more sophisticated and comprehensive consideration of PET in environmental studies, particularly those related to climate change.

*Code and data availability.* The Jupyter Notebook and data used in this study are available in the "examples" folder of the GitHub repository and also available on Zenodo (version v.1.3.1, DOI: 10.5281/zenodo.5896799). The authors welcome code contributions, bug reports, and feedback from the community to further improve the software. *PyEt* is free and open-source software available under the MIT license. Source code is available at the project's home page on GitHub. Full documentation is available on ReadTheDocs. *PyEt* is meant as a community project, and the Authors welcome contributions and feedback to continue to improve and develop the project are welcome.

## Appendix: PET Methods - Climate suitability and applications

Table A1: List of variables and symbols used in the paper and supplementary section

| Variable | Description | Units* |
|---|---|---|
| PET | Potential evapotranspiration | mm day$^{-1}$ |
| T | Mean daily temperature | °C |
| T$_{max}$ | Maximum daily temperature | °C |
| T$_{min}$ | Minimum daily temperature | °C |
| T$_{dew}$ | Mean daily dew-point temperature | °C |
| RH | Mean daily relative humidity | % |
| RH$_{max}$ | Maximum daily relative humidity | % |
| RH$_{min}$ | Minimum daily relative humidity | % |
| u$_2$ | Wind speed measured at 2 m | m s$^{-1}$ |
| R$_n$ | Net radiation | MJ m$^{-2}$ d$^{-1}$ |
| R$_s$ | Incoming solar radiation | MJ m$^{-2}$ d$^{-1}$ |
| R$_a$ | Extraterrestrial daily radiation | MJ m$^{-2}$ d$^{-1}$ |
| $G$ | Soil heat flux | MJ m$^{-2}$ d$^{-1}$ |
| n | Actual duration of sunshine | hour |
| N | Maximum possible duration of sunshine or daylight hours | hour |
| Elev | Elevation above sea level | m |
| lat | Latitude | radians |
| lat$_{deg}$ | Latitude | degrees |
| $p$ | Atmospheric pressure | kPa |
| $\lambda$ | Latent heat of vaporization | MJ kg$^{-1}$ |
| $\Delta$ | Slope of the saturation vapor pressure curve | kPa K$^{-1}$ |
| $\rho_a$ | Air density | kg m$^{-3}$ |
| $\rho_w$ | Water density (= 1) | Mg m$^{-3}$ |
| $c_p$ | Specific heat of dry air | MJ kg$^{-1}$ K$^{-1}$ |
| $e_0$ | Saturation vapor pressure of the air at T | kPa |
| $e_s$ | Saturation vapor pressure of the air | kPa |
| $e_a$ | Actual vapor pressure of the air | kPa |
| $\gamma$ | Psychrometric constant | kPa K$^{-1}$ |
| $r_s$ | Bulk surface resistance | s m$^{-1}$ |

| Variable | Description | Units* |
|---|---|---|
| $r_l$ | Bulk stomatal resistance | s m$^{-1}$ |
| $S_{r_l-[CO_2]}$ | Relative sensitivity of $r_l$ to $\Delta$ [CO$_2$] | ppm$^{-1}$ |
| $r_a$ | Bulk aerodynamic resistance | s m$^{-1}$ |
| h$_c$ | Crop height | m |
| LAI | Leaf area index | - |
| CO$_2$ | Atmospheric CO$_2$ concentration | ppm |
| K$_u$ | Unit conversion factor (=86400) | s d$^{-1}$ |
| a$_w$ | Penman wind coefficient | - |
| b$_w$ | Penman wind coefficient | - |
| C$_n$ | Numerator constant that changes with reference type | K mm s$^3$ Mg$^{-1}$ d$^{-1}$ |
| C$_d$ | Denominator constant that changes with reference type | s m$^{-1}$ |
| $\alpha$ | Surface albedo | - |
| $\alpha_L$ | Priestley-Taylor coefficient | - |
| P$_y$ | Percentage of actual day-light hours for the day compared to the number of day-light hour during the entire year | - |
| as1 | Empirical coefficient for extraterrestrial radiation | - |
| bs1 | Empirical coefficient for extraterrestrial radiation | - |
| a | Empirical coefficient for Net Long-Wave radiation | - |
| b | Empirical coefficient for Net Long-Wave radiation | - |
| k | Empirical/calibration coefficient | - |

**Table A2.** Overview of PET methods: climate suitability, applications, and limitations. $ET_0$ - reference crop/surface ET, $PET_C$ - potential crop/surface ET, $PET_{OW}$ - potential ET for open water, $PET_{SL}$ - potential ET for shallow lakes, $PET_{RR}$ - potential ET for rainfall-runoff modelling. Based on Allen et al. (1998); McMahon et al. (2013); Jensen and Allen (2016); Yang et al. (2021); Pimentel et al. (2023)

.

| Method name | Application | Climate | Limitations/Comments |
|---|---|---|---|
| Penman-Monteith | $ET_0$, $PET_C$, $PET_{RR}$ | All climates | High input data requirement |
| FAO-56 | $ET_0$ | All climates | High input data requirement |
| Penman | $PET_{OW}$, $PET_{SL}$, $PET_{RR}$ | All climates | High input data requirement |
| Priestley-Taylor | $PET_C$, $PET_{RR}$ | Temperate/polar | Calibration recommended for semi-arid and arid regions; often underestimates in high vapor pressure deficit areas. |
| Kimberly-Penman | $ET_0$-alfalfa | Temperate/continental | High input data requirement |
| Thom-Oliver | $ET_0$ | Temperate/continental | High input data requirement |
| Blaney–Criddle | $ET_0$, $PET_{RR}$ | Temperate | Overestimates in calm, moist, shaded areas; underestimates in windy, dry, sunny ones. |
| Hamon | $PET_C$ | All climates | Recommended regional calibration. |
| Romanenko | $PET_C$, $PET_{RR}$ | All climates | Best recommended model for PET in China. |
| Linacre | $PET_C$ | All climates | Recommended regional calibration. |
| Haude | $PET_C$ | All climates | Recommended regional calibration. |
| Turc | $ET_0$, $PET_C$ | Humid | Underestimates in areas with large daily vapor pressure deficits. |
| Jensen–Haise | $PET_0$, $PET_{RR}$ | Continental | Recommended regional calibration. |
| McGuinness–Bordne | $PET_C$, $PET_{RR}$ | All climates | Recommended regional calibration. |
| Hargreaves | $ET_0$, $PET_{RR}$ | Tropical/dry | Not recommended in windy or low $RH_{min}$ regions. May overestimate in humid climates. |
| FAO-24 | $ET_0$ | All climates | - |
| Abtew | $PET_C$ | Humid | Poor performance in arid climates. |
| Makkink | $PET_C$, $PET_{RR}$ | All climates | Originally designed for Western Europe, this method may underestimate higher PET. |
| Oudin | $PET_{RR}$ | All climates | Mainly used for hydrological modelling. |

## Appendix: PET Methods - equations

### Penman-Monteith (pm)

Through the introduction of the Penman-Monteith equation by Monteith (1965), a broad applicability of PET estimation to different surfaces and vegetation types was achieved (Jensen and Allen, 2016). This was done by implementing the plant aerodynamic resistance ($r_a$) and the surface resistance ($r_s$) in the PET formula:

$$PET = \frac{1}{\lambda \rho_w} \left[ \frac{\Delta(R_n - G) + \rho_a c_p K_u (e_s - e_a)/r_a}{\Delta + \gamma(1 + \frac{r_s}{r_a})} \right] \tag{A1}$$

Users of *PyEt* can include leaf/canopy cover measurements (Leaf Area Index - LAI) to calculate surface resistance ($r_s$), thereby accounting for the effects of crop management and phenology on PET. A modified stomatal resistance model also allows for the inclusion of the sensitivity of the stomatal resistance ($r_l$) to the atmospheric $CO_2$ concentration (as shown in, for example, Yang et al., 2019; Vremec et al., 2022):

$$r_s = \frac{r_l(CO_2)}{0.5 \text{LAI}} = \frac{r_{r_l - 300} \left\{ 1 + S_{r_l - CO_2}(CO_2 - 300) \right\}}{0.5 \text{LAI}} \tag{A2}$$

where $S_{r_l - [CO_2]}$ [ppm$^{-1}$] is the relative sensitivity of $r_l$ to $\Delta$ [$CO_2$] and $r_{r_l - 300}$ [s m$^{-1}$] is the reference stomatal resistance when atmospheric $CO_2$ concentration is 300 ppm. The relative sensitivity of $r_l$ to $\Delta$ [$CO_2$] represents the change in $r_l$ per ppm increase in $CO_2$ concentration.

If measurements of crop height exist, these data can be used to calculate the aerodynamic resistance to vapor and heat transfer ($r_a$) to represent the effects of crop phenology on PET:

$$r_a = \frac{\ln \left[ \frac{z_m - d}{z_{om}} \right] \ln \left[ \frac{z_h - d}{z_{oh}} \right]}{k^2 u_z} \tag{A3}$$

where $z_m$ is the reference level at which the wind speed is measured; $z_h$ is the height of the temperature and humidity measurements; $k$ is the von Karman constant ($= 0.41$), $u_z$ is the measured wind speed (Allen et al., 1998) and $d$ is the zero plane displacement height, taken as $0.67 h_c$ [m]; $z_{om}$ is the roughness parameter for momentum ($= 0.123 h_c$) [m] and $z_{oh}$ is the roughness parameter for heat and water vapor ($= 0.1 z_{om}$) [m] (Jensen and Allen, 2016).

Free parameters in the Penman-Monteith equation, available for calibration, include the bulk stomatal $r_l = 100$ (ranging between 40 - 150) and surface resistance (ranging between 50 - 200), with values for specific surfaces/crops found in (Jensen and Allen, 2016). Additionally, one can adjust the surface albedo ($\alpha = 0.23$, ranges between 0.04 for water surfaces and 0.9 for snow) (Jensen and Allen, 2016) to estimate net shortwave radiation (equation 38 in Allen et al. (1998)), or the empirical coefficients for net long-wave radiation $a = 1.35$ and $b = -0.35$ (equation 39 in Allen et al. (1998)), or the empirical coefficients for clear-sky radiation $as1 = 0.25$ and $bs1 = 0.5$ (equation 36 in Allen et al. (1998)). Optional meteorological inputs include G, $T_{max}$, $T_{min}$, $RH_{max}$, $RH_{min}$, $p$, N.

### ASCE-PM *(pm_asce)*

The ASCE Penman-Monteith equation for is computed after Walter et al. (2000) [equation 1]:

$$PET = \frac{0.408\Delta(R_n - G) + \gamma\frac{C_n}{T+273}u_2(e_s - e_a)}{\Delta + \gamma(C_d u_2)} \tag{A4}$$

where $C_n = 900$ and $C_d = 0.34$ are empirical coefficients for short reference vegetation (grass), while for tall reference vegetation (alfalfa) $C_n = 1600$ and $C_d = 0.38$ can be specified. The free parameters a, b, as1, bs1 and $\alpha$ are consistent with those specified for the Penman-Monteith method, while the optional meteorological inputs remain the same.

### FAO-56 *(pm_fao56)*

The FAO-56 Penman-Monteith equation for reference crop evapotranspiration is computed after Allen et al. (1998) [equation 6]:

$$PET = \frac{0.408\Delta(R_n - G) + \gamma\frac{900}{T+273}u_2(e_s - e_a)}{\Delta + \gamma(1 + 0.34u_2)} \tag{A5}$$

The free parameters a, b, as1, bs1 and $\alpha$ are consistent with those specified for the Penman-Monteith method, while the optional meteorological inputs remain the same.

### Penman *(penman)*

The Penman's PET formulation is computed after Penman (1948):

$$PET = \frac{1}{\lambda\rho_w}\left[\frac{\Delta(R_n - G) + \gamma(e_s - e_a)(a_w + b_w u_2)}{\Delta + \gamma}\right] \tag{A6}$$

where free parameters for the Penman's wind function include $a_w = 1$ and $b_w = 0.537$ (Valiantzas, 2006), while Penman (1948) suggested values of $a_w = 2.626$ and $b_w = 1.381$. The free parameters a, b, as1, bs1 and $\alpha$ are consistent with those specified for the Penman-Monteith method, while the optional meteorological inputs remain the same.

### Priestley-Taylor *(priestley_taylor)*

Priestley-Taylor's PET formulation is computed after Priestley and Taylor (1972):

$$PET = \alpha_L \frac{\Delta(R_n - G)}{\lambda\rho_w(\Delta + \gamma)} \tag{A7}$$

where $\alpha_L = 1.26$ is an empirical coefficient. The free parameters a, b, as1, bs1 and $\alpha$ are consistent with those specified for the Penman-Monteith method, while the optional meteorological inputs remain the same.

The Kimberly-Penman equation (Wright, 1982) is computed after Oudin et al. (2005):

$$PET = \frac{\Delta(R_n - G) + \gamma(e_s - e_a)w}{\lambda \rho_w (\Delta + \gamma)} \tag{A8}$$

where $w = u_2 \left[ 0.4 + 0.14 \exp\left( -\left( \frac{(j-173)}{58} \right)^2 \right) \right] + \left[ 0.605 + 0.345 \exp\left( -\left( \frac{j-243}{80} \right)^2 \right) \right]$.

The free parameters a, b, as1, bs1 and $\alpha$ are consistent with those specified for the Penman-Monteith method, while the optional meteorological inputs remain the same.

**Thom-Oliver** *(thom_oliver)*

Thom-Oliver's PET formulation is computed Thom and Oliver (1977), as used in Oudin et al. (2005):

$$PET = \frac{\Delta(R_n - G) + 2.5\gamma(e_s - e_a) \cdot a_w(1 + b_w u_2)}{\lambda rho_w (\Delta + \gamma(1 + \frac{r_s}{r_a}))} \tag{A9}$$

where $a_w = 2.6$ and $b_w = 0.536$. The free parameters a, b, as1, bs1, $\alpha$, $r_l$, $r_s$ and $S_{r_l - [CO_2]}$ are consistent with those specified for the Penman-Monteith method, while the optional meteorological inputs remain the same.

**Blaney-Criddle** *(blaney_criddle)*

Three different approaches can be taken to estimate the Blaney-Criddle PET, depending on the selected method:

$$PET = \begin{cases} a + bP_y(0.46T + 8.13) & \text{if } Method = 0 \text{ (after Schrödter (1985))}, \\ k \cdot P_y(0.457T + 8.128) & \text{if } Method = 1 \text{ (after Xu and Singh (2001) [equation 6])}, \\ k1 + b_{var} \cdot P_y(0.46T + 8.13) & \text{if } Method = 2 \text{ (after McMahon et al. (2013) [equation S9.7 and S9.8])}, \end{cases} \tag{A10}$$

where $k = 0.65$, $a = -1.55$ and $b = 0.96$ are empirical coefficients, and $k1 = 0.0043 \cdot RH_{min} - \frac{n}{N} - 1.41$, and $b_{var} = 0.81917 - 0.0040922 \cdot RH_{min} + 1.0705 \cdot \frac{n}{N} + 0.065649 \cdot u_2 - 0.0059684 \cdot RH_{min} \cdot \frac{n}{N} - 0.0005967 \cdot RH_{min} \cdot u_2$.

**Hamon** *(hamon)*

The PET formulation after (Hamon, 1963), as used in (Oudin et al., 2005):

$$PET = k \cdot \left( \frac{N}{12} \right)^2 \cdot \exp\left( \frac{T}{16} \right) \tag{A11}$$

where $k = 1$ is a calibration coefficient.

### Romanenko *(romanenko)*

Romanenko's PET formulation Romanenko (1961) as used in Oudin et al. (2005):

$$PET = k \left(1 + \frac{T}{25}\right)^2 \cdot \left(1 - \frac{e_a}{e_s}\right) \tag{A12}$$

where $k = 4.5$ is an empirical coefficient (Oudin et al., 2005). Optional meteorological inputs include $T_{max}$, $T_{min}$, $RH_{max}$ and $RH_{min}$.

### Linacre *(linacre)*

Linacre's PET formula Linacre (1977), as used in Oudin et al. (2005):

$$PET = \frac{\frac{500 \cdot T_m}{100 - \text{lat}_{\text{deg}}} + 15 \cdot (T - T_{\text{dew}})}{80 - T} \tag{A13}$$

where $T_m = T + 0.006 \cdot \text{Elev}$.

### Haude *(haude)*

Haude's PET formulation Haude (1955), as used in Schiff (1975) is computed as:

$$PET = k \cdot FK \cdot (e_0 - e_a) \cdot 10 \tag{A14}$$

where $k = 1$ is a calibration coefficient and $FK$ represents Haude's monthly coefficients, as adapted by Schiff (1975).

### Turc *(turc)*

The PET formula, as derived from Turc (1961) and used in McMahon et al. (2013) (equations S9.10 and S9.11), is computed as:

$$PET = k \cdot c \cdot \frac{T}{T + 15} (23.88 R_s + 50) \tag{A15}$$

where $k = 0.013$ is an empirical coefficient and $c$, dependent on the relative humidity (RH), is defined as:

$$c = \begin{cases} 1 + \frac{50 - \text{rh}}{70} & \text{if rh} < 50, \\ 1 & \text{otherwise.} \end{cases} \tag{A16}$$

**Jensen-Haise** *(jensen_haise)*

The PET according to the Jensen-Haise model (Jensen and Haise, 1963), varies depending on the chosen method:

$$PET = \begin{cases} k\frac{R_s}{\lambda\rho_w}(T - T_x) & \text{if } Method = 0 \text{ (after Jensen and Allen (2016))}, \\ k\frac{R_aT}{\lambda\rho_w} & \text{if } Method = 1 \text{ (after Oudin et al. (2005))}, \end{cases} \tag{A17}$$

where $cr = 0.025$ is an empirical coefficient and $T_x = -3$ as used in Jensen and Allen (2016).

**McGuinness-Bordne** *(mcguinness_bordne)*

McGuinness-Bordne's PET equation (McGuinness and Bordne, 1972), as used in Oudin et al. (2005):

$$PET = k\frac{R_a(T + 5)}{\lambda\rho_w} \tag{A18}$$

where $k = 0.0147$ is an empirical coefficient as suggested by Xu and Singh (2000).

**Hargreaves** *(hargreaves)*

The Hargreaves PET equation (Hargreaves and Samani, 1982) is computed as:

$$PET = kc_{HS}\frac{R_a}{\lambda\rho_w}\sqrt{T_{\max} - T_{\min}}(T + 17.8) \tag{A19}$$

- If *method = 0* (after Jensen and Allen (2016)), the empirical coefficient $k = 0.0135$, and $c_{HS} = 0$.

- If *method = 1* (after McMahon et al. (2013)), $k = 0.0135$. The coefficient $c_{HS}$ is calculated as $0.00185 \cdot (T_{\max} - T_{\min})^2 - 0.0433 \cdot (T_{\max} - T_{\min}) + 0.4023$.

**FAO-24** *(fao_24)*

The FAO-24 PET equation from Doorenbos (1977); Jensen et al. (1990), as used in Xu and Singh (2000) (equation 11 and 12), is given by:

$$PET = a + \frac{\Delta}{\Delta + \gamma}\frac{R_s}{\lambda\rho_w}(1 - \alpha)b \tag{A20}$$

where $a = -0.3$ and $b = 1.066 - 0.13 \cdot \frac{RH}{100} + 0.045u_2 - 0.02 \cdot \frac{RH}{100}u_2 - 0.315 \cdot \left(\frac{RH}{100}\right)^2 - 0.0011u_2$. Free parameters include the surface albedo $\alpha$, consistent with those used in the Penman-Monteith method.

**Abtew** *(abtew)*

Abtew's PET equation (Abtew, 1996), as used in Xu and Singh (2000) (equation 14):

$$PET = k\frac{R_s}{\lambda\rho_w} \tag{A21}$$

where $k = 0.53$ is an empirical coefficient as suggested by Xu and Singh (2000).

**Makkink** *(makink)*

Makkink's PET equation (Makkink, 1957):

$$PET = \frac{\Delta}{\Delta + \gamma}\frac{R_s}{\lambda\rho_w} \tag{A22}$$

where $k = 0.65$ is the empirical coefficients recommended by Hiemstra and Sluiter (2011) ranging between (0.61-0.77) Jensen and Allen (2016).

**Makkink-KNMI** *(makink_knmi)*

The Royal Netherlands Meteorological Institute (KNMI) employs a slightly modified version of the Makkink equation, tailored specifically for conditions in the Netherlands, as described by Hiemstra and Sluiter (2011):

$$PET = k\frac{\Delta}{\Delta + \gamma}\frac{R_s}{\lambda\rho_w} \tag{A23}$$

where $k = 0.65$. The calculations for $s$, $e_s$, $\gamma$, and $\lambda$ are specified as follows:

- $\Delta = \frac{7.5 \cdot 237.3}{(237.3+T)^2}\ln(10) \cdot e_s$,

- $e_s = 0.6107 \cdot 10^{\frac{7.5T}{237.3+T}}$,

- $\gamma = 0.0646 + 0.00006T$,

- $\lambda = (2501 - 2.375T)1000$.

**Oudin** *(oudin)*

According to Oudin et al. (2005) (equation 2), the potential evapotranspiration (PET) can be expressed as:

$$PET = \begin{cases} R_a\frac{(T+k_2)}{\lambda\rho_w k_1} & \text{if } T + k_2 > 0 \\ 0 & \text{otherwise,} \end{cases} \tag{A24}$$

where $k_2 = 5$ and $k_1 = 100$ (ranging between 75-100) are empirical coefficients recommended by Oudin et al. (2005).

## A1  Performance metrics

This appendix provides mathematical definitions of the performance metrics used to evaluate the PET models discussed in the manuscript. The model bias (mm/day) is calculated as the average difference between the PET values estimated using *PyEt* ($\hat{y}_i$) and the reference PET values, which include (i) literature values presented in Figure 1, and (ii) PET values computed using the Penman-Monteith method in Example 3 ($y_i$), over $n$ time steps:

$$\text{Bias} = \frac{1}{n}\sum_{i=1}^{n}(\hat{y}_i - y_i) \tag{A25}$$

The coefficient of determination ($R^2$) was computed as follows:

$$R^2 = 1 - \frac{\sum_{i=1}^{n}(y_i - \hat{y}_i)^2}{\sum_{i=1}^{n}(y_i - \overline{y})^2} \tag{A26}$$

where $\overline{y}$ is the mean of the reference data.

The Kling-Gupta Efficiency (KGE) is defined as:

$$\text{KGE} = 1 - \sqrt{(r-1)^2 + (\alpha-1)^2 + (\beta-1)^2} \tag{A27}$$

where $r$ is the correlation coefficient between the reference and estimated data, $\alpha$ is the ratio of the standard deviation of estimated data to that of reference data, and $\beta$ is the ratio of the mean of estimated data to that of reference data (bias ratio).

*Author contributions.* Conceptualization, M.V., S.B. and R.C.; software, M.V. and R.C.; investigation, M.V.; writing—original draft preparation, M.V. and R.C. ; writing—review and editing, S.B. and R.C.; supervision, S.B.. All authors have read and agreed to the published version of the manuscript.

*Competing interests.* The authors declare no conflict of interest.

*Acknowledgements.* We acknowledge the financial support by the University of Graz and the funding of the Earth System Sciences research program of the Austrian Academy of Sciences (ÖAW project ClimGrassHydro). We acknowledge the ZAMG dataset (https://data.hub.zamg.ac.at), KNMI dataset (https://www.knmi.nl/home), and the E-OBS dataset from the EU-FP6 project UERRA (http://www.uerra.eu) and the Copernicus Climate Change Service, and the data providers in the ECA&D project (https://www.ecad.eu). We appreciate the insightful comments and suggestions from two anonymous reviewers and the handling editor that greatly improved this manuscript. We also thank the contributors from the preprint discussions in HESS and the PyEt community for their valuable input.

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
