# Peer review of "PyEt v1.3.1: a Python package for the estimation of potential evapotranspiration"

_Geoscientific Model Development, 2024_

## Author Response (AR3)

We thank the reviewers for their valuable and constructive comments on our manuscript. We have considered each comment and revised the manuscript accordingly. We respond to each comment below. In the following, the reviewers' comments are in black font and our replies are in blue font.
* * *
Reviewer #1:

This paper presents a python package to implement twenty different PET methods. This package is going to be quite useful in hydrological modeling. The paper is quite well written and is easy to read. Therefore, I don't have any major suggestions. There are a few things that the authors can add to further improve the paper:

Thank you for your encouraging feedback and comments. We have responded to your comments below.

Add the equations of each of the 20 models in the paper or in the Appendix. Importantly, specify the units of each of the variables along with that of PET. I see that the units have been provided on GitHub, but providing the units to the paper would also be useful.

We have included all equations and their corresponding units into the Appendix. Each variable and the PET itself are now clearly defined with their respective units.

Include the plausible ranges of ET model parameters that can be treated as free parameters. This will be very helpful for the modelers who are new to these models.

Based on the literature, plausible ranges for the free parameters exist for only a few PET methods. We've included the suggested parameter values next to the equations in the Appendix and provided their plausible ranges where available from the references.

Re-check the performance measures in the Figure 5. The R2 does not change between default and calibrated scenarios. In the lines 236, you mention that R2 increases. But it does not.

We reviewed the performance measures in Figure 5 and corrected the text in line 236 to accurately state that $R^2$ remains unchanged between the default and calibrated scenarios. This is because the calibration, involving a uniform adjustment of all values by the same factor (calibration coefficient), does not significantly enhance the linear correlation between the observed and simulated data. While this adjustment improves the KGE by better aligning the mean and variability, it does not affect the linear relationship captured by $R^2$.

Reviewer #2:

This paper presents a Python package for Potential Evapotranspiration (PET) estimation, integrating 20 different methods within a single framework. I appreciate the effort to consolidate these methods into one publicly available platform, facilitating an understanding of various PET estimation approaches for users, which aids practical applications. However, the paper lacks novel scientific contributions or innovations, particularly given the existence of R packages for PET estimation and numerous established publications in this area.

Thank you for your feedback and constructive comments. We regret that the manuscript did not effectively convey the importance of this work to the scientific community, but are confident that our revised version improves on this. Your individual comments are replied to below.

Specific comments:

Although the authors acknowledge the uncertainties associated with different PET estimation methods and the additional uncertainties introduced by climate change, no uncertainty analysis is provided. The Penman-Monteith method is regarded as the standard. If one method is indeed the standard, the necessity of including other methods must be clearly justified.

We recognize that our initial wording was unclear, as we did not mean that the PM method can be used as a standard or benchmark for the other methods. We have revised the text to state: 'Apart from the Penman-Monteith method, which is considered the standard by the Food and Agriculture Organization (FAO) and the World Meteorological Organization (WMO), multiple alternative methods are also available in PyEt.'  Moreover, to avoid confusion we removed section with the formulas from the PM method, and moved it to the appendices along with the equations from all other PET methods (see also reply to comment from reviewer 1).

While the FAO and WMO acknowledge the Penman-Monteith method as the standard, we advocate the use (and hence, justify the inclusion in PyEt) of multiple methods for the following reasons (added text in paragraph 3 and 4 in the introduction):

· Data Availability: The Penman-Monteith method often requires extensive data inputs, which may not always be available. Alternative methods with fewer data requirements are necessary to ensure PET estimation is feasible in data-scarce regions/periods (example 2 for data-scarce regions and example 3 for data-scarce periods).

· Uncertainty Analysis: Using multiple methods can enhance our understanding of the uncertainties associated with PET estimation. This approach helps in capturing a range of potential outcomes, thereby providing more robust and reliable predictions (example 1 and 3, supported by studies such as Zhou et al., 2020; Dakhlaoui et al., 2020; Yang et al., 2019, Bormann, 2010; Seiller and Anctil, 2016; Gharbia et al., 2018; Shi et al., 2020).

These points underscore the importance of offering a variety of methods in PyEt, ensuring that users can select the most appropriate method based on data availability and the need for comprehensive uncertainty analysis.

While acknowledging the Penman-Monteith method as a standard reference, the manuscript should objectively evaluate the potential value of other empirical approaches, especially for applications in data-sparse regions or for quantifying uncertainties across an ensemble of methods.

Examples 1, 2, and 3 in the manuscript address this comment. Example 2 demonstrates how alternative empirical approaches, such as the Hargreaves method, can be beneficial in regions where the necessary data for the Penman-Monteith method is unavailable. Example 3 illustrates how empirical methods can be calibrated against the Penman-Monteith method when data is available and then used for hindcasting in data-sparse periods. These examples plus the references of Zhou et al., (2020); Dakhlaoui et al., (2020); Yang et al., (2019); Bormann, (2010); Seiller and Anctil, (2016); Gharbia et al.,(2018); Shi et al., (2020); highlight the value of alternative methods in expanding the applicability of PET estimation across data scarce spatial/temporal periods and in enhancing uncertainty quantification.

The introduction lacks a compelling motivation for the research. It should clearly identify key remaining scientific gaps or problems in PET estimation that PyET aims to solve through new research or technical capabilities.

We recognize the need for a clearer motivation in our research. We have rewritten the introduction to outline why PyEt is needed, particularly paragraph 3,4, and 5 in the introduction. We refer to the change document where these changes are highlighted. We believe this revision effectively identifies the key scientific gaps and problems in PET estimation that PyET aims to address through its new research and technical capabilities.

A detailed comparison with existing R packages for PET estimation is essential. Highlight the significant advantages of your Python package over these existing tools. Clarify the scientific contributions of PyET beyond merely including more methods.

A comparison of the computed PET values from different methods with the R package is provided in Figure 1. Significant differences of our Python package are highlighted in the introduction (lines 54-62) and the conclusion. Specifically, we emphasize PyEt's flexibility in handling both time series and gridded data, its integration with widely used Python libraries, and its ability to process large datasets efficiently. Additionally, PyEt adheres to FAIR principles for research software, enhancing findability, accessibility, interoperability, and reusability, which are crucial for advancing reproducible research. We also apply best software development practices, including full documentation, testing using continuous integration, git-versioning, and a community platform (GitHub) where all code (changes) are traceable and issues can be tracked.

The model description section should provide detailed information on the included PET methods, provide an in-depth analysis of the fundamental assumptions, limitations, and suitability of each PET approach for different hydroclimatic regimes and data availabilities. Clearly articulate the significant advantages offered by PyET in terms of new scientific insights, functionality, performance, or other technical merits.

We addressed the reviewer's comments by making the following enhancements:

- Detailed Information on PET Methods: All equations and their free parameters have been added to the Appendix for comprehensive reference.

- Analysis of assumptions, limitations, and suitability: Table A2 is included to provide an in-depth analysis of the fundamental assumptions, limitations, and suitability of each PET approach for different hydroclimatic regimes and data availabilities.

- Significant Advantages of PyEt: The significant advantages offered by PyEt, such as its flexibility in handling both time series and gridded data, improved performance through integration with Python libraries, and adherence to FAIR principles, have been clearly articulated in the introduction (lines 54-70) and conclusion.

These additions aim to clarify the scientific contributions of PyEt beyond merely including more methods, highlighting its functionality, performance, broad adoption, and other technical merits.

Editor #

In Figure 5, there are three metrics including bias, R-squared, and KGE used. What is KGE? Kling-Gupta Efficiency? It was not mentioned within the text. Mention it within your methodology with clear citation of the method (see https://doi.org/10.1016/j.jhydrol.2009.08.003). If possible, within the manuscript such as in appendix, also include the formulae for R-squared, and KGE.

Include in your discussion guidance regarding comprehensive manner of evaluating the performance of various PET computation methods.

Thank you for your comments and suggestions concerning the performance metrics used in our manuscript. We have now included a detailed description of the Kling-Gupta Efficiency (KGE) in the methods section, with the appropriate citation. Additionally, we have added the equations for R2 and KGE to the appendix. We have also expanded our discussion section to include guidance on evaluating the performance of various PET methods. We appreciate your guidance in enhancing the quality of our work.